GRID-independent molecular descriptor analysis and molecular docking studies to mimic the binding hypothesis of γ-aminobutyric acid transporter 1 (GAT1) inhibitors

Zafar Sadia
Jabeen Ishrat ishrat.jabeen@rcms.nust.edu.pk
Research Center for Modeling and Simulation (RCMS), National University of Sciences and Technology (NUST) , Islamabad, Federal , Pakistan
Rocha Joao
Electronic publication date: 2019 Jan 31
Publication date: 2019
Volume: 7
Electronic Location ID: e6283
Received 2018 Oct 11; Accepted 2018 Dec 14
Copyright: © 2019 Zafar and Jabeen
Copyright year: 2019
Copyright holder: Zafar and Jabeen
License: This is an open access article distributed under the terms of the Creative Commons Attribution License, which permits unrestricted use, distribution, reproduction and adaptation in any medium and for any purpose provided that it is properly attributed. For attribution, the original author(s), title, publication source (PeerJ) and either DOI or URL of the article must be cited.
License URL: https://creativecommons.org/licenses/by/4.0/

Keywords: 3D QSAR, GRIND model, GABA transporter 1 (GAT1), Tiagabine, hGAT1 inhibitors, Docking, Structure–activity relationship (SAR)

Funding: HEC ‘Indigenous Ph.D. Fellowship for 5,000 scholars’ Phase-II, Batch-I, 2012 Support was provided by HEC ‘Indigenous Ph.D. Fellowship for 5,000 scholars’ Phase-II, Batch-I, 2012. The funders had no role in study design, data collection and analysis, decision to publish, or preparation of the manuscript.

==============================
Background

The γ-aminobutyric acid (GABA) transporter GAT1 is involved in GABA transport across the biological membrane in and out of the synaptic cleft. The efficiency of this Na+ coupled GABA transport is regulated by an electrochemical gradient, which is directed inward under normal conditions. However, in certain pathophysiological situations, including strong depolarization or an imbalance in ion homeostasis, the GABA influx into the cytoplasm is increased by re-uptake transport mechanism. This mechanism may lead to extra removal of extracellular GABA which results in numerous neurological disorders such as epilepsy. Thus, small molecule inhibitors of GABA re-uptake may enhance GABA activity at the synaptic clefts.

Methods

In the present study, various GRID-independent molecular descriptor (GRIND) models have been developed to shed light on the 3D structural features of human GAT1 (hGAT1) inhibitors using nipecotic acid and N-diarylalkenyl piperidine analogs. Further, a binding hypothesis has been developed for the selected GAT1 antagonists by molecular docking inside the binding cavity of hGAT1 homology model.

Results

Our results indicate that two hydrogen bond acceptors, one hydrogen bond donor and one hydrophobic region at certain distances from each other play an important role in achieving high inhibitory potency against hGAT1. Our docking results elucidate the importance of the COOH group in hGAT1 antagonists by considering substitution of the COOH group with an isoxazol ring in compound 37, which subsequently leads to a three order of magnitude decrease in biological activity of 37 (IC50 = 38 µM) as compared to compound 1 (IC50 = 0.040 µM).

Discussion

Our docking results are strengthened by the structure activity relationship of the data series as well as by GRIND models, thus providing a significant structural basis for understanding the binding of antagonists, which may be useful for guiding the design of hGAT1 inhibitors.

Introduction

Brain functioning is controlled by neuron circuits that release excitatory and inhibitory neurotransmitters like glutamate and γ-aminobutyric acid (GABA) and neuromodulators like norepinephrine, dopamine and serotonin (Heng, Moonen & Nguyen, 2007). The removal of neurotransmitters from the extracellular space (i.e., between pre- and post- synaptic neurons) is regulated by their respective transporters (Calapai et al., 2001). However, imbalances in the levels of these neurotransmitters at synaptic clefts which may be associated with several neurological disorders, including Alzheimer’s disease, schizophrenia, Parkinson’s disease, anxiety, sleep disorders and epilepsy (Sherin & Nemeroff, 2011; Pirttimaki, Parri & Crunelli, 2013; Shetty & Bates, 2015). Most of these neurological disorders are associated with the GABAergic system and are mainly modulated using allosteric agonists of the GABAA receptor. However, inhibition of the GABA re-uptake transport to maintain its concentration gradient at synaptic clefts represents a promising concept for treating neurological disorders (Carvill et al., 2015).

GABA transporters (GATs) are categorized into four subtypes, GAT1-3, and betaine/GABA transporter 1 (BGT1) (Jin et al., 2011). GAT1 and 3 are mainly expressed at GABAergic neurons and glial cells, respectively, throughout the brain (Minelli et al., 1995; Minelli et al., 1996; Conti et al., 1998; Melone, Ciappelloni & Conti, 2015). However, GAT2 is localized at arachnoid and ependymal cells and has very low expression in neurons and glial cells in the brain (Conti et al., 1999; Jin et al., 2011); BGT1 is expressed, from low to high concentration, in the liver, kidney, meninges and at the blood brain barrier (Anderson, Kidd & Eskandari, 2010). All GATs belong to the neurotransmitter sodium symporter family. These transporters use a sodium gradient for re-uptake of the neurotransmitters out of the synaptic cleft; however, in certain cases a reverse transport mode is also known, which releases the neurotransmitter in a nonvesicular way (Yu et al., 1998). Overall, 75% of GABA re-uptake is mediated by GAT1 (Parpura & Haydon, 2008; Zafar & Jabeen, 2018). This reflects that GAT1 is mainly accountable for GABA transport and related disorders. Therefore, development of potential antagonists of this transporter to maintain the concentration gradient of GABA at synaptic clefts may represent a potential therapeutic strategy. Up to now, Tiagabine is the only second-generation FDA approved anticonvulsant agent that selectively inhibits Homo sapiens GAT1 (hGAT1). However, Tiagabine analogs that have been developed are often associated with off target toxicity and poor ADME-Tox properties that lead to side effects such as sedation, tremors and ataxia (Madsen et al., 2011). Thus, developing new chemical scaffolds of GABA reuptake inhibitors (i.e., hGAT1 antagonists) that have maximum efficacy and reduced toxicity might aid in the successful treatment of neuronal disorders.

Previously, various antagonists of hGAT1, including nipecotic acid, guvacine, proline, pyrrolidine, azetidine and THPO derivatives (Dalby, 2000; Andersen et al., 2001; Clausen et al., 2005; Fülep et al., 2006; Faust et al., 2010; Hellenbrand et al., 2016; Schmidt, Höfner & Wanner, 2017; Lutz et al., 2018; Tóth, Höfner & Wanner, 2018), have been synthesized and pharmacologically tested and optimized using structure–activity relationship (SAR) data. Additionally, several ligand-based strategies including 2D QSAR (Jurik et al., 2013), CoMFA (Zheng et al., 2006) and pharmacophore models (Hirayama, Díez-Sampedro & Wright, 2001; Nowaczyk et al., 2018) have been developed to optimize small molecule inhibitors against hGAT1. However, most of these studies were class specific, focusing on nipecotic acid derivatives (Petrera et al., 2015), Tiagabine analogs (Jurik et al., 2015) and triarylnipecotic acid derivatives (Dhar et al., 1994). Recently, a nipecotic acid derivative DDPM-2571 has been synthesized with one log unit greater inhibitory potency against GAT1 as compared to Tiagabine which showed anticonvulsant, antidepressant and antinociceptive effects in mouse models (Sałat et al., 2017). Moreover, a novel class of allosteric GAT1 antagonists has been identified through mass spectrometry screening of pseudostatic hydrazone libraries. Hauke et al. (2018) suggested that the identified allosteric nipecotic acid derivatives may provide physiological relevance in terms of hGAT1 modulation as their interaction in hGAT1 binding pocket differs from Tiagabine. Additionally, some reports also suggest 5-aminolevulinic acid (5-Ala) may also inhibit the cellular uptake of GABA by GAT isoforms (Rud et al., 2000). Until very recent, no X-ray crystal structure of any hGAT has been published. Therefore, various hGAT1 models in different conformations have been developed previously using the crystal structure of the leucine transporter (LeuT) from Aquifex aeolicus (PDB ID: 3F3A) as a template. These models may assist to study the binding of hGAT1 antagonists and to study the ion dependent transport mechanistic of GABA through hGAT1 (Bicho & Grewer, 2005; Jurik et al., 2015).

In the present study, we aim to develop predictive GRID-independent molecular descriptor (GRIND) models to provide deeper insight into the 3D structural features of hGAT1 antagonists. Moreover, a recently published X-ray structure of dopamine transporter (DAT) in Drosophila melanogaster (dDAT, PDB ID: 4XP4, resolution: 2.8 Å, sequence identity: 46%) (Wang, Penmatsa & Gouaux, 2015b) is used in the current study to build a model of hGAT1, followed by molecular docking studies to probe how nipecotic acid and N-diarylalkenyl piperidine analogs bind to the binding cavity of hGAT1.

Methods

Dataset

A complete workflow of hGAT1 antagonists data pre-processing and cleaning has been provided in Fig. 1. Briefly, a dataset of 580 hGAT1 antagonists, along with their respective binding affinities (IC50) ranging from 0.04 to 8511 µM, was obtained from the literature (Dhar et al., 1994; Schousboe, 2000; Clausen et al., 2005, 2006; Fülep et al., 2006; Zheng et al., 2006; Alexander, Mathie & Peters, 2007; Reith, 2007; Faust et al., 2010; Alexander, Mathie & Peters, 2011; Nakada et al., 2013; Quandt, Höfner & Wanner, 2013; Sitka et al., 2013). Subsequently, duplicates and fragments were removed from the data, followed by the removal of antagonists with a molecular mass less than 150 and IC50 >100 µM. The duplicate antagonists were the replicated chemical compounds with biological activities determined through different biological assays including [3H] GABA uptake assay, GAT1 transport assay, radio-ligand binding assay and equilibrium binding assay using different expression systems like Xenopus oocytes and HEK cell lines (Dhar et al., 1994; Kragler, Höfner & Wanner, 2008; Nakada et al., 2013). Moreover, the antagonists with molecular mass less than 150 were excluded from the analysis because they were representing molecular fragments and therefore may not be selective against the hGAT1. Similarly, antagonists with IC50 > 100 μM were also discarded as they reflect least active compounds in comparison with the most active antagonist of the database (IC50 = 0.040 μM). Thus, only the data with IC50 values ranging from 0.040 to 100 μM was used for further analysis. Overall, our data comprises of total 215 antagonists that were further subjected to biological cleaning by selecting only those antagonists whose IC50 values were evaluated using a radio labeled [3H] GABA uptake assay (152 antagonists) in hGAT1 expressing HEK cells (102 antagonists). Furthermore, 3D structures of the selected 102 antagonists (Table S1) were constructed and energy minimized using the MMFF94x force field (Halgren, 1996) in MOE version 2013.0802 (Chemical Computing Group, 2013). The final data set of the 102 antagonists of hGAT1 consists of nipecotic acid, proline, pyrrolidine, exo-THPO and N-diarylalkenyl piperidine derivatives that follow a general pattern of attachment of a COOH group at the ortho, meta or para positions of piperidine, proline, pyrrolidine, or azetidine rings, followed by a linker region of variable lengths. The general architecture of hGAT1 antagonists in the present data set is represented in Fig. S1. It has been reported previously that attachment of aromatic moieties to the linker region is highly correlated with the activity of hGAT1 antagonists (Faust et al., 2010). Depending on the type of cyclic moieties attached at the linker region, the present data set of hGAT1 antagonists was divided into three main classes (classes A, B and C). Overall, the whole data set of 102 hGAT1 inhibitors in Table S1 was further divided into a training set (80%) and a test set (20%) by using the diverse subset selection method (Schmuker, Givehchi & Schneider, 2004; Gillet, 2011). Briefly, 300 2D as well as 3D descriptors available in MOE version 2013.0802 (Chemical Computing Group, 2013) were computed to obtain a distance calculation for each database entry. 20% of the data structures (19 compounds) with larger distance values from each other were selected as the test set and the remaining 83 compounds (80%) were used to train the model (Minh, Klaere & von Haeseler, 2009) using GRIND (Durán & Pastor, 2011). Additionally, a recently published data set of 15 nipecotic acid derivatives was used as validation set (Lutz et al., 2017).

Figure 1 Workflow of the hGAT1 antagonists data pre-processing and cleaning.

GRID-independent molecular descriptor

It has been previously reported that GRIND variables are highly dependent on the 3D conformations of molecules (Durán & Pastor, 2011). Therefore, multiple conformational search approaches were used to generate four different 3D conformational sets of the training data. Subsequently, each independent sets of molecular conformations, including energy minimized, extended 3D, induced fit docking (IFD) solutions, and molecular alignment conformations by pharmacophore mapping approach of hGAT1 antagonists along with their −log IC50 values, were independently loaded into the software package Pentacle v 1.06 (Durán & Pastor, 2011) for the construction of four different GRIND models. Various authors have demonstrated that hGAT1 antagonists bind in their protonated state (Jurik et al., 2015). Therefore, all the compounds in the different conformational sets were protonated at a physiological pH of 7.4. A complete protocol to train the 3D QSAR model using GRIND is provided in Fig. S2. Briefly, includes following steps: Computation of the molecular interaction fields (MIFs): MIF computation was done using four different probes including DRY (hydrophobic probe), TIP (steric hot-spot defining molecular shape), N1 (hydrogen bond acceptor) and O (hydrogen bond donor) within a molecule. The total energy at each node was calculated as a sum of the Lennard-Jones potential (Elj) (Lennard-Jones, 1931; Bouanich, 1992), the electrostatic energy (Eel), and the hydrogen bond energies (Ehb) by iteratively placing each probe at different GRID steps: Exyz=∑Elj+∑Eel+∑Ehb

Discretization: AMANDA algorithm (Durán, Martínez & Pastor, 2008) was used to discretize the MIFs using default energy cutoff values of −0.75, −0.5, −4.2 and −2.6 for the TIP, DRY, N1 and O probes, respectively, to pre-filter the nodes that fails to meet the energy cutoffs.

Encoding: The consistently large auto and cross correlation algorithm (Durán & Pastor, 2011) with default parameters was used for the encoding of the pre-filtered nodes at each discretization step. This produces consistent sets of variables whose values are directly represented in the form of correlogram plots. To correlate the structural variance of the training set data with respective biological activity (−log IC50), a partial least square (PLS) analysis has been performed using the full set of 570 active GRIND variables for each model. Moreover, to remove the inconsistent set of variables, cycles of fractional factorial design (FFD) (Baroni et al., 1993) were employed to obtain a good statistical model.

Multiple conformational analysis

Standard 3D conformations

Software CORINA v4.1.0 (Sadowski, 2003) was used to generate standard 3D conformations of the molecules. Briefly, CORINA build the 3D model of a molecule by combining the monocentric fragments with standard bond lengths, bond angles and dihedral angles, including the torsion angles of ring systems for their proper closure and to minimize the non-bonded (Van der Waals and electrostatic) interactions that occur within the flexible parts of the molecules (Sadowski, Schwab & Gasteiger, 2004). Finally, generated standard 3D conformation of each hGAT1 antagonist in the data set was further subjected to Pentacle v 1.06 (Durán & Pastor, 2011) for GRIND analysis.

Energy minimized conformations

The stochastic search algorithm in MOE v2013.0802 (Chemical Computing Group, 2013) was used to produce energy minimized conformations of the hGAT1 antagonists. A total of 250 conformations were generated and ranked according to their energy values. Each individual in the data set with the lowest energy score was selected for the GRIND analysis.

Induced fit docking (IFD)

The IFD (Sherman et al., 2006) protocol in MOE v2013.0802 (Chemical Computing Group, 2013) was followed to generate the docking solutions of 102 hGAT1 antagonists within the binding pocket of the hGAT1 model. The binding cavity was selected to include the residues already known to be involved in ligand–protein interactions in hGAT1, i.e., G59, Y60, A61, I62, G63, G65, N66, W68, Y86, L136, Y139, Y140, I143, Q291, F294, S295, Y296, G297, L300, N327, S328, S331, A358, L392, D395, S396 and D451 (Zhou, Zomot & Kanner, 2006; Skovstrup et al., 2010; Jurik et al., 2015). Overall, 20 poses per ligand were generated using default scoring function (London dG) and placement method (Triangle Matcher). Finally, the best scored docking conformation for each ligand was selected for further GRIND analysis.

Pharmacophore mapping approach

Another set of molecular conformations was generated by the pharmacophore mapping approach (Martin et al., 1993) using MOE v2013.0802 (Chemical Computing Group, 2013). Briefly, the standard 3D conformation of the prototype ligand Tiagabine was used as a template for the flexible alignment of the rest of the data. The best scored aligned system based on the energy values, shown in Fig. 2, was selected to develop the GRIND model.

Figure 2 Best aligned conformations of complete data set (102 compounds) of hGAT1 antagonists on standard 3D conformation of Tiagabine as a template.

Overall, the relevance of structural properties to the binding affinity has been determined by PLS analysis. However, the value of the correlation coefficient (R2) for the external test set validation was determined through Eq. 1 as described by Alexander, Tropsha & Winkler (2015).

(1) R2=∑(y−y^)2∑(y−y¯)2

Where y represents the experimentally determined biological activity (−log IC50) of the data set, y¯ is its mean value and y^ represent the corresponding predicted biological activity (−log IC50) by the GRIND model. The highly predictive final GRIND model helps to identify the 3D structural features of hGAT1 antagonists. However, in order to get deeper insight into the ligand interaction profiles within the hGAT1 binding site, further structure-based studies have been performed.

Homology modeling

Due to the absence of a crystal structure of hGAT1, comparative modeling was performed using the recently solved crystal structure of the DAT in D. melanogaster (dDAT, resolution: 2.80 Å, PDB ID: 4XP4) (Wang, Penmatsa & Gouaux, 2015a). Comparative modeling of hGAT1 was performed using Modeller 9v8 software (Šali et al., 1995) Briefly, the primary sequence of hGAT1 (P30531) was retrieved from the UniProt KB databank (Magrane, 2011). Homologous transporter proteins include the LeuT and the DAT in the bacteria A. aeolicus (Yamashita et al., 2005) and D. melanogaster, respectively (Wang, Penmatsa & Gouaux, 2015a). Therefore, the homologous protein dDAT (PDB ID: 4XP4) in an open-to-out conformation was selected as a template for multiple sequence alignment (Schrödinger, 2017) because it shares 46% sequence identity with hGAT1 as identified by BLASTp algorithm against the Protein Data Bank database (https://blast.ncbi.nlm.nih.gov/), compared to AaLeuT, which only shares 25% sequence identity with hGAT1. Additionally, dDAT also contains a Cl− ion, which is necessary to maintain the ionic concentration gradient in mammalian GABA transport (Wang, Penmatsa & Gouaux, 2015b). In contrast, in AaLeuT, the Cl− ion was not co-crystallized.

Flanking regions of the N- and C-terminus of hGAT1 were removed from the hGAT1 model built via Modeller 9v8 (Šali et al., 1995). However, no residues were trimmed from the extracellular loop 2 (EL2) of hGAT1, as was done in previously reported studies, due to the difference in the number of amino acid residues in EL2 for AaLeuT and hGAT1 (Baglo et al., 2013). The sodium ions and a chloride ion in a stoichiometry of 2:1 were added at the positions seen in dDAT. Briefly, 100 models of hGAT1 were built and the quality of these models was evaluated through ERRAT (Colovos & Yeates, 1993), PROCHECK (Lovell et al., 2003) and Verify3D (Bowie, Luthy & Eisenberg, 1991; Lüthy, Bowie & Eisenberg, 1992) using the web server at http://servicesn.mbi.ucla.edu/. The final model was energy minimized using the MMFF94x force field of LigX (Labute, 2008) in MOE v2013.08 (Halgren, 1996). Finally, the model quality was again evaluated using a Ramachandran plot (Ho & Brasseur, 2005).

Docking and pose analysis of selected hGAT1 modulators

Molecular docking was performed to get deeper insights into the binding of hGAT1 antagonists using GOLD suite v5.2.2 (Jones, Willett & Glen, 1995). To remove any bias in the pose generation step, different rotamers of the side chains were sampled. However, the ligand binding site was kept flexible. The binding site was sampled by keeping a distance of 17 Å around the amino acid residues G59, Y60, A61, I62, G63, G65, N66, W68, Y86, L136, Y139, Y140, I143, Q291, F294, S295, Y296, G297, L300, N327, S328, S331, A358, L392, D395, S396 and D451. These amino acid residues are already known from mutagenesis data to be important in interactions within hGAT1 (Zhou, Zomot & Kanner, 2006; Skovstrup et al., 2010; Jurik et al., 2015). Additionally, 2 Na+ ions and a Cl− ion that are known for their role in the GABA transport mechanism were also incorporated into the binding cavity (Zhou, Zomot & Kanner, 2006; Skovstrup et al., 2010; Jurik et al., 2015). The side chains of hGAT1 (with the exception of ligand binding site residues) were kept rigid while the antagonists were treated flexible by performing 100 genetic algorithm runs per molecule using the gold score fitness function (Jones, Willett & Glen, 1995). The best pose for each ligand inside the hGAT1 binding pocket was selected for further ligand–protein interaction analysis.

To illustrate the binding hypothesis for hGAT1 antagonists with chemically different scaffolds, the docking solutions of selected compounds from each class, including compounds 2, 6, 8, and 15 from class A, compounds 14, 27, 36, and 40 from class B and compounds 1, 3, 4, 37, and 84 from class C were used for further hierarchical clustering analysis based on the root mean square deviation (RMSD) of the common scaffold (Stanton et al., 1999) of each respective antagonist class. The compound selection criteria were solely based on the SAR data, as explained in the ‘Results’ section. A complete docking and pose selection protocol is provided in Fig. S3. Overall, a total of 172, 110 and 64 clusters were obtained for classes A, B and C, respectively, at 2 Å on the basis of the RMSD of the heavy atoms of the common scaffold (Stanton et al., 1999). Final cluster of binding poses from each class was further selected on the basis of SAR data as explained in results section.

Results

GRID-independent molecular descriptor (GRIND) analysis

The statistical parameters for each GRIND model developed from different conformational set are shown in Table S2. Unfortunately, none of the four conformational sets of ligands produced a statistically good model with the full set of variables. Thus, FFD (Baroni et al., 1993) was applied to reduce the number of inconsistent variables in each model. After the first cycle of FFD, the statistical parameters of each model were slightly improved. However, a final model with good statistical parameters (q2 = 0.59, r2 = 0.75 and SDEP = 0.44) (Table S2) was obtained after two cycles of FFD with the pharmacophore-based aligned set of conformations on Tiagabine template.

Multiple linear regression analysis using leave one out cross validation (Elisseeff & Pontil, 2003) of the final model resulted in a plot of observed versus predicted inhibitory potencies (−log IC50) of the training data, as shown in Fig. 3. All the compounds in the training set (filled circles), test set (hollow circles) and validation set (triangles) are well-predicted, with a difference of less than one log unit between the observed and the predicted biological activity values with the exception of compound 35, for which the predicted activity (−log IC50) value is 1.52 log units greater as compared to experimental activity value (Fig. 3). However, no outlier has been identified in the test set (R2: 0.53, Table S3) and the validation set (R2: 0.57). Briefly, the higher predicted activity value for compound 35 (actual −log IC50/predicted −log IC50: −1.44/0.08) (outlier in Fig. 3) is may be due to its high lipophilicity (logP = 7.80) which may lead to poor bioavailability and thus, overall poor experimental efficacy.

Figure 3 Correlation plot between experimental versus predicted inhibitory potencies (−log IC50) of hGAT1 antagonists.

Training set, test set and validation set are represented with filled circles, hollow circles and triangles, respectively. The chemical structure represents observed outlier (compound 35) in the training set.

Figure 4A shows PLS coefficients correlograms of the final GRIND model. Positive and negative peaks in auto and cross correlograms of different variables (O, N1, DRY and TIP) elucidate their positive or negative contributions, respectively, towards the inhibitory potency (−log IC50) against hGAT1. Additionally, it depicts the 3D structural features and their mutual distances that best describe their roles in the inhibition of hGAT1. It is evident from the PLS co-efficient correlogram that the O–O, N1–N1, DRY–N1, O–N1 and TIP–TIP variables have major influences on the inhibitory potency of the data set.

Figure 4 (A) PLS coefficients correlograms showing the descriptors directly (positive value) or inversely (negative values) correlated to −log IC50 values of the dataset.

A change in biological activity against hGAT1 is depicted by N1–N1, O–N1, DRY–N1, TIP–TIP and O–O variables. (B) N1–N1 probes (blue contours) represents two hydrogen bond acceptor groups (OH and carbonyl group of COOH) at a mutual distance of 8.0–8.40 Å within highly active ligands (0.049–0.75 µM) whereas (C) represent N1–N1 pair of probes at a distance of 14.00–14.40 Å in least active compounds (5.00–38 µM) (D) reflects O–N1 pair of variable depicting a hydrogen bond donor (O: red contour) and a hydrogen bond acceptor (N1: blue contour) at a mutual distance of 5.60–6.00 Å within the highly active molecules (0.049–0.75 µM) while (E) represents O–N1 variable contours at a distance of 10.40–10.80 Å from each other in least active compounds (5.00–38 µM) (F) represent DRY–N1 pair of probes delineating a hydrophobic (DRY: yellow contour) at a distance of 10.40–10.80 Å from a hydrogen bond acceptor region (N1: blue contour) in the active ligands (0.049–0.75 µM) (G) represent DRY–N1 pair of probe at distance of 6.40–6.80 Å in least active compounds (4.34–78 µM) (H, I) represent O–O variables depicting two hydrogen bond donor probes (O: red contours) at a mutual distance of 6.00–6.40 Å within the molecules; (J) TIP–TIP pair of variable represent two molecular boundaries (green contours) at a mutual distance of 12.40–12.80 Å.

N1–N1 variables in Fig. 4A depict the presence of two hydrogen bond acceptor groups within the molecules that are present at a mutual distance of 8.00–8.40 Å in highly active (0.049–0.75 µM) antagonists of hGAT1. In the current data set, it represents a distance between the carbonyl and hydroxyl groups of the COOH attached at the ortho, meta and para positions of the piperidine ring in the nipecotic acid derivatives, as shown in Fig. 4B. All the compounds in the present data set (classes A, B and C) possess COOH groups attached at the ortho, meta or para positions of the piperidine ring. Compound 37 in class C is the only exception, where the COOH group has been substituted with an isoxazol ring. The N1–N1 correlogram for compound 37 represents a distance of 8.00–8.40 Å between the tertiary nitrogen of the piperidine ring and the carbonyl group of the isoxazol ring. Additionally, the N1–N1 pair of variables at a longer distance range of 14.00–14.40 Å shows a negative contribution towards the hGAT1 inhibition potential and, thus, has been identified in the least active (5.00–38 µM) compounds in the data set. Briefly, it represents a distance between the carbonyl oxygen of the COOH group and the ether group present either in the linker region or in the bulky aromatic substituents, as shown in Fig. 4C.

Interestingly, a sharp positive peak in the O–N1 correlogram in Fig. 4A demonstrates that a hydrogen bond acceptor at a distance of 5.60–6.00 Å from a hydrogen bond donor region has a positive effect on the inhibitory potency against hGAT1. Within the present data set, the hydrogen bond donor region represents the protonated nitrogen of the piperidine ring, while the COOH group provides the hydrogen bond acceptor region, as shown in Fig. 4D. This distance has been identified in the most active (0.049–0.75 µM) hGAT1 antagonists and is absent in the least active (5.00–38 µM) antagonists. This further strengthens the importance of the carbonyl group and the protonated nitrogen for the high inhibitory potency of hGAT1 antagonists. Additionally, the O–N1 pair of probes at a larger distance range of 10.40–10.80 Å, as shown in Fig. 4E, has been identified in the least active (6.56–78 µM) compounds within the data series and, thus, has a negative effect on the overall inhibitory potency against hGAT1. In the present data series, this represents a distance between the protonated nitrogen (hydrogen bond donor) of the piperidine ring and the methoxy substitution of the diaryl moieties (hydrogen bond acceptor), as shown in Fig. 4E. Previously, the importance of the protonated nitrogen group in the piperidine ring of hGAT1 antagonists has been demonstrated by Zheng et al. (2006) by CoMFA analysis of N-diarylalkenyl-piperidinecarboxylic acid analogs. Thus, it is tempting to speculate that the methoxy groups at the N-diarylalkenyl rings in the data series may have a negative effect on the inhibitory potency against hGAT1.

Similarly, the DRY–N1 pair of probes in Fig. 4A represents one hydrophobic region (DRY) and one hydrogen bond acceptor variable region within the molecules (N1). Both features have been identified at a mutual distance of 10.40–10.80 Å in highly active compounds (0.049–0.75 µM) and at a distance of 6.40–6.80 Å in the least active compounds (4.34–78 µM). In highly active compounds, this represents the distance between aromatic moieties (thiophene rings, benzene rings and tricyclic rings) and the COOH group at the piperidine, proline, pyrrolidine, or azetidine ring, as shown in Fig. 4F. However, in the least active compounds it represents a distance between the aromatic moieties and the ether group in the linker region, as shown in Fig. 4G. Interestingly, all the selected compounds of classes A, B and C follow the distance pattern of highly active compounds in the DRY–N1 correlogram. Moreover, compound 37 (IC50 = 38 µM) of class C shows a distance of 6.40–6.80 Å between the carbonyl group of the isoxazol ring and the benzene ring substituent at the linker region, which may provide a reason for the low biological activity of compound 37.

Furthermore, the O–O correlogram in Fig. 4A shows the presence of two hydrogen bond donors at a mutual distance of 6.00–6.40 Å within the least active (10–78 µM) compounds and, thus, may contribute negatively towards the inhibitory potency of the data set against hGAT1. For instance, in compound 32 this distance is associated with the ether group present in the hydrophilic chain and the para-monofluoro groups attached to the aromatic rings, as shown in Fig. 4H. Thus, it may be inferred that attachment of any electronegative atom to the ortho or para positions of the aromatic moieties is not favorable for achieving high biological activity values in this data series. Similarly, another example of the O–O correlogram is presented by compound 36 of class B, for which this variable represents the distance between the ether linker region of the hydrophilic chain and the para-methoxy group attached at the R2 group, as shown in Fig. 4I.

Furthermore, the TIP–TIP correlogram in Fig. 4A represents two molecular boundaries of methoxy substitutions on the aromatic moieties at a mutual distance of 12.40–12.80 Å in the least active compounds (11.00–78 µM). In the present data set, this represents the distance between the para-methoxy substituents at R1 and R2 of the aromatic rings in class B (Fig. 4J) and the distance between benzene ring attached at the linker region and the isoxazol ring in compound 37 of class C (IC50 = 38 µM). Interestingly, the results of the TIP–TIP correlogram are further strengthened by our findings for the O–O correlogram that addition of para-methoxy groups to the aromatic moieties results in reduced biological activity. Interestingly, N1-TIP correlogram in Fig. 4A represents a distance of 18.00–18.40 Å between the molecular boundary depicted by the R1 and R2 substitutions of the aromatic rings and the COOH group attached to the piperidine ring in the least active compounds. Thus, it may suggest the negative impact of R1 and R2 substitutions towards hGAT1 inhibitory potency as the importance of COOH group is already evident from N1–N1 and O–N1 variables. Overall, a brief summary of all the specific probes contributing towards the activity of hGAT1 antagonists is presented in Table 1.

Table 1 Summary of GRIND variables and their corresponding distances identified as highly correlated to biological activity (−logIC50) of compounds.

Probes	Distances (Å)	Features	Impact	Comments	
N1–N1	8.00–8.40	OH
C=O	+	COOH group at meta position of piperidine, pyrrolidine, or azetidine ring has shown positive contribution towards hGAT1 inhibition activity (IC50)	
14.00–14.40	C=O
–O–	−	Carbonyl oxygen of the COOH group and the ether group present either in the linker region or in the bulky aromatic substituents has shown negative contribution towards the biological activity.	
O–N1	5.60–6.00		+	Distance between protonated nitrogen of the piperidine, pyrrolidine, or azetidine ring and the COOH group	
10.40–10.80		−	Protonated nitrogen of the piperidine ring and the methoxy substitution of the diaryl moieties	
DRY–N1	10.40–10.80	Di/tri aryl moieties
COOH	+	Distance between COOH group of piperidine, proline, pyrrolidine, or azetidine ring and bulky aromatic rings after linker chain	
6.40–6.80	Di/tri aryl moieties
–O–	−	Distance between aromatic moieties and the ether group in the linker region	
O–O	6.00–6.40	–O–
X atom (any electronegative atom e.g., F, Cl−, O−, F−)	−	Depicts a distance between the ether group of hydrophilic chain and methoxy or flouro group attached at para position of aromatic rings	
TIP–TIP	12.40–12.80	OCH3
OCH3	−	Distance between the methoxy substitutions on aromatic moieties attached at the linker region of hGAT1 antagonists	

External validation of the final GRIND model

To further demonstrate the predictive ability of the trained GRIND model, a recently published data set of 15 nipecotic acid derivatives possessing an alkyne type spacer (linker region) bridging the polar region at the aromatic rings, as shown in Table 2, was used for external validation (Lutz et al., 2017).

Table 2 External validation set of nipecotic acid derivatives of hGAT1 inhibitors.

	
Validation set compound # (VSC)	n	R1	R2	IC50 µM	−log IC50	Predicted −log IC50	Residual value	
VSC_1	1	H	H	0.10	1.00	0.24	0.75	
VSC_2	1	F	H	0.07	1.12	0.33	0.78	
VSC_3	1	CH3	CH3	0.20	0.68	−0.06	0.74	
VSC_4	1	CH3	Cl	0.15	0.80	0.49	0.30	
VSC_5	2	Cl	H	1.58	−0.20	0.33	−0.53	
VSC_6	2	F	H	2.75	−0.44	0.26	−0.70	
VSC_7	2	CH3	H	1.28	−0.11	0.13	−0.24	
VSC_8	2	H	Cl	3.38	−0.53	−0.04	−0.48	
VSC_9	2	Cl	Cl	0.93	0.03	−0.19	0.22	
VSC_10	2	F	F	1.99	−0.30	−0.38	0.08	
VSC_11	2	Cl	F	0.97	0.01	−0.06	0.07	
VSC_12	2	CH3	Cl	0.97	0.01	0.20	−0.19	
VSC_13	2	CH3	F	1.31	−0.12	0.19	−0.31	
VSC_14	2	H	H	3.80	−0.58	0.37	−0.95	
VSC_15	2	CH3	CH3	3.46	−0.54	0.40	−0.94	

Overall, the IC50 values of the data set range between 0.07 and 3.80 µM (Table 2). All the biological testing results (inhibitory potencies) were the mean of three independent experiments ± the standard error of the mean (Lutz et al., 2017). Remarkably, all the compounds in the external validation set are well-predicted with R2 value of 0.57 (Fig. 3, represented by triangles), and exhibiting a difference of less than one log unit between the experimental and predicted inhibitory values, as shown in Table 2.

Homology modeling of hGAT1

Based on the alignment (representing 66% sequence similarity (Fig. S4); equivalent to 46% sequence identity) hGAT1 models in open to out conformation were constructed using dDAT as a template as explained in methodology section. The Ramachandran plot of the final selected model with best ERRAT value (81.28) in comparison to ERRAT values of rest of the models (range: 69.93–80.00) and Verify3D score (83.24%) has shown four residues in the outlier region, including F174 (EL2), N176 (EL2), R419 (TM9), and V528 (TM11). In order to further improve the model quality, final selected model was energy minimized via LigX using MMFF94x in MOE v2013.0802 (Halgren, 1996; Chemical Computing Group, 2013). After the energy minimization, the model was re-evaluated by PROCHECK and none of the residues were observed in the disallowed region (Fig. S5). The Ramachandran plot displayed 93.2% of the residues of hGAT1 in most favored regions, 5.9% residues in additionally allowed regions, 0.9% residues in generously allowed regions, and 0.0% residues in disallowed regions. Similarly, the ERRAT (87.92) and Verify3D (85%) scores were also improved compared to the model before energy minimization. This further shows the reliability of the final model.

Briefly, ERRAT score provides information about the residues that contribute towards the lower percentage/score of the hGAT1 model (i.e., residues 165 to 180). This is may be because EL2 in hGAT1 contains a greater number of residues than in AaLeuT and dDAT. These may be the flanking residues of this loop. Furthermore, Gether et al. (2006) reported that binding of the ligand to the transporter in the open-to-out conformation results in the penetration of extracellular loop 4 (EL4) deep inside the hGAT1 binding site (i.e., TM1 residues), leading to the formation of a lid-type structure that seals the ligand into the hydrophobic core/region of the binding cavity. This confirms that the binding of the substrate to the transporter in its open-to-out form involves a major conformational change in EL4 (Gether et al., 2006). In the constructed hGAT1 model, EL4 penetration into the hydrophobic region of the binding cavity has resulted in the adjustment of bi- or tri-aromatic/ cyclic moieties of hGAT1 antagonists. Overall, the final model has 12 transmembrane (TM) segments, two Na+ ions and one Cl− ion along with the bridging extracellular and intracellular loops. One side of the hydrophobic cavity around cyclic moieties is defined by the residues W68 (TM1b), Y139 (TM3), Y140 (TM3) and I143 (TM3) and the other by F294 (TM6a), S359 (EL4a), D451 (TM10), and S456 (TM10). The final structural model of hGAT1 in the open-to-out conformation, along with the two Na+, one Cl− and binding pocket residues are shown in Fig. S6.

Molecular docking of selected ligands in hGAT1

Structure–activity relationship

On the basis of the SAR, four compounds each from classes A and B and five compounds from class C have been selected for further structure-based studies, as shown in Table 3.

Table 3 Selected compounds from classes A, B and C for experimental guided docking studies.

	
Compound #	Class scaffold	R1	R2	R3	IC50 (μM)	
2	A	–CH3	–CH3	m–COOH (R)	0.049	
6	A	–CH2OC6H5	–CH2OC6H5	m–COOH (R)	0.34	
8	A	–CH2OC6H5	–CH2OC6H5	o–COOH(R,S)	0.65	
15	A	–CH2OC6H5	–CH2OC6H5	p–COOH(R,S)	1.5	
14	B	H	H	H	1.4	
27	B	H	–OCH3	H	6.9	
36	B	–OCH3	–OCH3	–OCH3	30	
40	B	–OCH3	–OCH3	H	43	
1	C	–ON=C(C6H5)2	–	–	0.040	
3	C	–CH=C(C6H5)2	–	–	0.20	
4	C	–OCH(C6H5–m–CF3)2	–	–	0.26	
37	C		38	
84	C		0.11	

Briefly, class A includes N-diarylalkenyl piperidine COOH derivatives (Table S1) with a COOH group substituent at the ortho, meta or para position of the piperidine ring. COOH group in compounds 2 and 6 was R-configured however, stereochemistry of rest of the compounds in class A was not specified and thus, represent racemic mixtures. Compound 2 (Tiagabine, IC50 = 0.049 µM), the only FDA approved antiepileptic drug, was selected for molecular docking to investigate its binding hypothesis and to compare it with rest of the analogs in class A. Compound 6 (IC50 = 0.34 µM) was selected to evaluate the one order of magnitude decrease in its inhibitory potency compared to compound 2, which may be due to the presence of bulky substituents (–CH2OC6H5) at R1 and R2 in compound 6 compared to methyl substituents at the same positions in compound 2. Interestingly, compounds 8 and 15 have the same bulky substituents (–CH2OC6H5) at R1 and R2 as compound 6; however, an approximate two-fold decrease in the biological activity has been observed, i.e., 6 (IC50 = 0.34 µM) > 8 (IC50 = 0.65 µM) > 15 (IC50 = 1.5 µM) that can be correlated to the position (meta > ortho > para) of the COOH group on the piperidine ring. Briefly, compound 6 (IC50 = 0.34 µM), which has the COOH group at the meta position (m–COOH) of the piperidine ring, showed two-fold greater inhibitory potency against hGAT1 compared to compound 8 (IC50 = 0.65 µM), which has a COOH group at the ortho position, and in turn showed two-fold greater activity as compared to compound 15 (IC50 = 1.5 µM), which has a COOH group at the para position (p-COOH). Therefore, compounds 6, 8 and 15 were also selected for docking within the binding cavity of hGAT1 to understand the binding hypothesis of the COOH group at the ortho, meta and para positions of the piperidine ring.

Class B of hGAT1 antagonists consists of ethyl trityl ether derivatives of nipecotic acid that possess a R- configured COOH group at the meta position (mCOOH) of the piperidine ring and R1, R2 and R3 methoxy substituents on the triphenylmethyl group, as shown in Table S1. Compound 14 being the highly active in class B (IC50 = 1.40 µM), was selected for comparison with the rest of the data. Attachment of a para-methoxy group at R2 in compound 27 results in a 5-fold decrease in the biological activity (6.9 µM) compared to compound 14. Interestingly, para-methoxy substituents at R1, R2 and R3 in compound 36 (IC50 = 30 µM) and at R1 and R2 in compound 40 (IC50 = 43 µM) result in a decrease of approximately two orders of magnitude in the inhibitory potency against hGAT1 compared to compound 14. This suggests that the decrease in the biological activity of compounds 27, 36 and 40 compared to compound 14 is may be due to the presence of para-methoxy groups at R1, R2 and R3, which might cause steric hindrance within the binding pocket or limit the access of the compounds to the hGAT1 binding pocket.

This is further validated by our findings from the GRIND model that para-methoxy substituents at aromatic moieties and the ether linker group at a mutual distance of 6.00–6.40 Å (O–O probes) have a negative effect on the hGAT1 inhibitory potency. Therefore, compounds 27, 36 and 40 were also selected in addition to compound 14 for further molecular docking studies to probe the effect of para-methoxy substitutions on aromatic moieties on binding within the active site of hGAT1.

Compounds 1, 3, 4, 37 and 84, which are diaryl derivatives of nipecotic acid (all having racemic COOH groups with the exception of compound 37 as shown in Table S1) were selected from class C to elucidate the effect of the attachment of diaryl and the piperizine derivatives at the linker region on the biological activity of hGAT1 antagonists. Compound 1 (NNC-711), the 2-benzhydrylideneamino derivative of nipecotic acid, was selected as a reference ligand in the class since it is a known selective inhibitor of hGAT1 (IC50 = 0.040 µM) (Kragler, Höfner & Wanner, 2005). However, replacement of the –C=NO– in compound 1 with –C=CH– and –CHO– in compounds 3 (SK&F-100330A) and 4 (CI-966) resulted in one order of magnitude decrease in the biological activities of compounds 3 (IC50 = 0.20 µM) and 4 (IC50 = 0.26 µM) compared to 1 (IC50 = 0.040 µM). This decrease in biological activity has been correlated to an increase in the logP values, logP 4 (5.45) > logP 3 (4.19) > logP 1 (3.79), which may reflect the importance of polar groups in interactions within the binding pocket of hGAT1. As a result, compounds 1, 3 and 4 were selected for further molecular docking and ligand–protein interaction profiling. Additionally, compound 84 was selected to compare its ligand–protein interaction profile with compound 3. In compound 37, replacement of the COOH group with 4,5,6,7-tetrahydroisoxazolo[4,5-c]pyridin-3-ol resulted in a three order of magnitude decrease in the inhibitory potency of the compound (IC50 = 38 µM) compared to compound 1 (IC50 = 0.040 µM). Therefore, compound 37 was selected to elucidate the effect of nipecotic acid on the inhibitory potency against hGAT1. This may point towards the importance of nipecotic acid in obtaining high inhibitory potency against hGAT1.

Briefly, all the selected compounds from class A (2, 6, 8, 15), class B (14, 27, 36, 40) and class C (1, 3, 4, 37, 84) were docked into the binding site of the hGAT1 model as explained in the methods section. To remove any biases in the docking protocol, 100 poses per ligand were generated using the GOLD score fitness function. However, the fitness function in Gold score is optimized for the prediction of binding positions rather than binding affinities (Jones, Willett & Glen, 1995) therefore, a poor correlation (R2: 0.02) has been observed between biological activity (−logIC50) and top scored pose of each ligand (Fig. S7; Table S4). Therefore, in order to remove any biases in the pose selection criteria, for the final ligand–protein interaction analysis, only one cluster exhibiting maximum docked ligands from each class was selected on the basis of the SAR and mutagenesis data, followed by energy minimization of the final ligand–protein complexes (Fig. S7; Table S4) (Halgren, 1996; Chemical Computing Group, 2013).

Briefly, a total number of 14 clusters of docking solutions of compounds in class A, 12 clusters of binding conformations of compounds in class B, and four clusters of binding solutions for the compounds in class C have been identified that contained the maximum number of docked ligands. Briefly, all 14 clusters of class A contained three (2, 8, 15) out of four docked ligands, 12 clusters of class B also contained three (14, 27, 36) out of four docked ligands and four clusters of class C contained four (1, 3, 37, 84) out of five docked ligands as shown in Fig. 5.

Figure 5 Docking poses of clusters from classes A, B and C in hGAT1 binding pocket that contained maximum number of docked ligands from respective classes.

(A) 14 clusters from class A containing three (2, 8, 15) out of four ligands. (B) 12 clusters from class B exhibiting three (14, 27, 36) out of four ligands. (C) Four clusters from class C possessing four (1, 3, 37, 84) out of five ligands.

The binding position of all the 14 clusters in class A was approximately the same. However, the binding conformations of the docking solutions in each cluster were different. Therefore, the interaction patterns of all 14 clusters of class A were explored and only one cluster that is strengthened by the SAR and mutagenesis data was selected for the final ligand–protein interaction analysis. A similar selection procedure was repeated for the 12 clusters of binding solutions for the selected compounds in class B and the four clusters of binding poses for the selected compounds in class C. Sodium ion (represented by Na1) was involved in interactions with the COOH attached to the piperidine ring in all three classes, as well as with the amino acid residues A61, N66, S295, and N327, as shown in Fig. 6A.

Figure 6 Winning clusters obtained from docking of hGAT1 antagonists.

(A) Final docking poses of hGAT1 antagonists from classes A, B and C. Positioning of di- or tri-aromatic rings represent the hydrophobic areas and charged piperidine ring represent the polar region in the binding pocket of hGAT1. (B) Binding positions of winning clusters of classes A, B and C in hGAT1 binding pocket along with Na1, Na2 (blue spheres) and a Cl− (green sphere) ion.

Overall, the winning cluster of class A showed interactions with TM segments 1a, 1b, 6a, 6b and 10 (Fig. 6B). It has been observed that the carbonyl group at the meta position of the piperidine ring in compound 2 shows a strong hydrogen bonding interaction with the –NH group of G65, whereas the thiophene rings of the common scaffold shows a π–π interaction with Y140 at TM3 and W68 at TM1b (Fig. 7A). Additionally, F294 of TM6a shows a strong hydrogen bonding interaction with the protonated nitrogen of the piperidine ring. In compounds 8 and 15 hydrogen bonding is observed between the –NH group of G65 and the OH group of the COOH, as shown in Fig. 7B. Thus, the one order of magnitude decrease in the inhibitory potency of compounds 8 and 15 compared to compound 2 may be attributed to a decrease in the hydrogen bonding strength due to a shift in the position of the hydrogen bond from the carbonyl group (in compound 2) to the hydroxyl group (in compounds 8 and 15). The COOH flipping is might be due to the shift of the COOH group from the meta position to the ortho and para positions of the piperidine ring in compounds 8 and 15, respectively.

Figure 7 Interaction pattern of binding solutions of classes A, B and C in hGAT1 model.

Ligand–protein interactions of (A) compound 2 and (B) compound 8 of class A. Interactions pattern of (C) compound 14 and (D) compound 36 of class B. Binding poses of (E) compound 1, (F) compound 3 and (G) compound 37 of class C.

Additionally, the ligand–protein interaction pattern of the final docking poses of compounds 8 and 15 reveals that bulky substituents at the R1 and R2 positions are more exposed to the extracellular environment (Fig. 8). This disrupts the π-π stacking between the phenyl ring of Y140 and the thiophene ring associated with R1 in compounds 8 and 15, which was present in compound 2. However, the π–π interaction between the thiophene ring of the common scaffold of class A and the indole ring of W68 at TM1b and the hydrogen bonding interaction between the protonated nitrogen of the piperidine ring and F294 at TM6a remained intact, as shown in Fig. 7B. Therefore, this shows that the bulky substituents on the thiophene rings in compounds 8 and 15 result in a loss of the interaction with Y140, which is known to be critical for the transport activity of GATs (Bismuth, Kavanaugh & Kanner, 1997). These results agree with our SAR data showing that the attachment of bulky groups at the R1 and R2 positions in class A, together with the positioning of the COOH on the piperidine ring, may affect the inhibitory potency against hGAT1. Overall, the amino acid residues G65, W68, Y140 and F294 are involved in interactions with the N-diarylalkenyl piperidine COOH derivatives within the hGAT1 binding cavity.

Figure 8 Outward projection of bulky substitutions on aromatic moieties of antagonists of class A and B from the hGAT1 binding pocket.

Similarly, out of 12 clusters of binding solutions of compounds in class B, only one cluster (Fig. 6B), which contains the binding poses of compounds 14, 27 and 36, followed the SAR pattern in their ligand–protein interaction profile. However, the rest of the clusters shared similar overall binding positions within the binding pocket of hGAT1. As all the compounds in class B have a meta-COOH and a protonated nitrogen in piperidine ring, their binding solutions showed a very similar pattern of hydrogen bonding interactions with the residues G65 and F294 and with Na1, as shown by compound 2 in class A. However, the benzene ring of the common scaffold in compound 14 shows a π–π interaction with the indole ring of W68 at TM1b (Fig. 7C). Additionally, both compounds 27 and 36 (Fig. 7D) shows hydrogen bonding interactions between the ether group in the linker region and the pyrrole ring of W68. Similar to the compounds 8 and 15 in class A, the methoxy substitutions on the aromatic moieties in compounds 27 and 36 projected out of the hGAT1 binding pocket (Fig. 8). They may require a large space in the cavity compared to compound 14, and as a result showed no interaction with the residues in the hGAT1 binding pocket.

Similar to the selected clusters of class A and B, the final cluster of class C followed the SAR and mutagenesis data. The final binding solutions for class C contain compounds 1, 3, 37 and 84, which are located in the vicinity of TM1a, TM1b, TM4, TM6a and TM6b (Fig. 6B). The selected compounds in class C show a π–π interaction with W68, followed by a similar hydrogen bonding interaction pattern at the protonated nitrogen atom (NH–O=C-F294) and at the COOH group (C=O–HN-G65) (Figs. 7E and 7F) (ligplot shown in Fig. S8) as that discussed for classes A (compound 2) and B. However, compound 37 is an exception in which the COOH is replaced with a five membered isoxazol ring. The residues surrounding Na1 are similar in the case of compound 37, as shown in Fig. 7G. Thus, the two to three orders of magnitude decrease in the inhibitory potency of compound 37 compared to its congeners is might be due to lack of COOH group and its associated interactions within the hGAT1 pocket.

Discussion

In present study, we demonstrated the importance of COOH group towards hGAT1 inhibition. Our final GRIND model mapped the distances of important pharmacophoric features of the ligands, including one hydrogen bond acceptor (N1), one hydrogen bond donor (O) and one hydrophobic (Dry) probe from the COOH group at meta- position of the piperidine ring as shown in Fig. 9 which reflect that COOH group at meta position of the piperidine ring may provide an important interaction point within the binding cavity of hGAT1.

Figure 9 Important hotspots regions for the high inhibitory potency of hGAT1 inhibitors.

One hydrogen bond acceptor contour (N1) at a distance of 8.00–8.40 Å from second hydrogen bond acceptor (N1) group, at a distance of 5.60–6.00 Å from a hydrogen bond donor (O) and at a distance of 10.40–10.80 Å from a hydrophobic (DRY) group.

Figure 9 is further validated by our SAR and docking protocol, where all three classes A, B and C showed interactions between the protonated nitrogen atom and F294 and between the COOH group and G65. The effect of the COOH group, as well as the protonated nitrogen atom, on the inhibitory potency of hGAT1 modulators has been previously demonstrated by various authors (Zhao et al., 2005; Fülep et al., 2006; Kragler, Höfner & Wanner, 2008; Pizzi et al., 2011). Therefore, three orders of magnitude decrease in inhibitory potency of compound 37 as compared to compound 1 is may be due to absence of COOH group in its chemical scaffold.

Furthermore, COOH groups at the meta position of docked antagonists (class A; compound 2 and class B) has been observed in the equatorial conformation with respect to Na1 in the hGAT1 model (Fig. 6A). Thus, it suggests that the meta positioning of the equatorial COOH group is more favorable for obtaining more potent hGAT1 antagonists compared to equatorial ortho or para COOH positioning. This observation agrees with previous studies by Skovstrup et al. (2010) and Jurik et al. (2015), who demonstrated that the R-configured COOH group in an axial conformation with respect to Na1 is involved in intramolecular hydrogen bonding with the protonated nitrogen of the piperidine ring.Thus, attachment of bulky groups (i.e., COOH) in an axial configuration with respect to Na1 is not favorable (Jurik et al., 2015). Previously, various authors proposed R-configured COOH group in hGAT1 antagonists as favorable binding conformation (Borden et al., 1994; Wermuth, 2008; Schmidt, Höfner & Wanner, 2017). However, in present study the stereochemical effect of hGAT1 antagonists in ligand protein interaction could not be considered due to lack of complete stereoisomers data.

Our results also demonstrate the negative impact of methoxy substitutions at R1, R2 or R3 positions of di- or tri-aryl rings and the ether linker group towards GAT1 inhibition. We hypothesize that 7-fold decrease in the biological activity of compound 27 (IC50 = 6.9 µM) and one order of magnitude decrease in the biological activity of compound 36 (IC50 = 30 µM) as compared to compound 14 (IC50 = 1.4 µM) is may be due to attachment of methoxy groups at R1, R2 or R3. This is supported by the O–O (Figs. 4H and 4I) and TIP–TIP correlograms (Fig. 4J) of the final GRIND model. Both distance features (O–O and TIP–TIP) are arising from methoxy or ether linker substituents and show a negative effect on the inhibitory potency against hGAT1. In compounds 27, 36 and 40 of class B, these map the distance between the ether in the linker region and the para-methoxy groups at the R2 position. Hence the biological activity values of compounds 27, 36 and 40 (6.9 µM, 30 µM and 43 µM) are significantly reduced compared to compound 14 (1.4 µM), which lacks electronegative atom substituents at the R1, R2 and R3 positions. Additionally, two orders of magnitude decrease in the inhibitory potency of compounds in class B compared to compounds in class A is may be attributed to the presence of the para-methoxy group in the common scaffold for class B compared to common scaffold of class A. This is further strengthened by a study by Pizzi et al. (2011), who studied the effect of ortho, meta and para substitutions on 4,4-diphenylbut-3-enyl derivatives and observed that biological activity against hGAT1 reduces with the substitution of methyl, chloride, fluoride and bromide at the ortho position of the di-aryl rings. This is also strengthened by the ligand protein interaction analysis of the final docking poses of compounds of classes A, B and C which elucidate that the bulky substitutions at R1 and R2 positions of di- or tri-aryl rings are projected out of the binding cavity, exposed to the extracellular environment and thus, are not well fitted within the binding cavity of hGAT1.

Overall, the ligand–protein interactions profile of selected hGAT1 inhibitors of calss A, B and C showed significant role of G65, W68, Y140 and F294 amino acid residues within the hGAT1 binding pocket. Previously, Baglo et al. (2013), revealed hydrogen bonding and π-π stacking with G65, N66, Y140, F294 and S295by docking substrate-like small molecules (i.e., 5-aminolevulinic acid (ALA), and methyl ester of ALA (MAL)) in the hGAT1 model. Additionally, various structure-based studies identified the binding hypothesis of Tiagabine and demonstrated the role of the amino acid residues W68, Y139, Y140, I143, F294, A358 and S359 in the formation of hydrophobic pockets in hGAT1 (Skovstrup et al., 2010; Jurik et al., 2015). Overall, the ligand–protein interaction profiles of the final binding solutions of hGAT1 inhibitors from classes A, B and C were compared to the already known interaction patterns of other classes of respective modulators, as shown in Fig. S9. It is evident from Fig. S9 that the N-diarylalkenyl piperidine derivatives of class A show an overlap with all of the residues already reported in the literature except D451. Similarly, the interaction profile of class B agrees with that of class A, with the exception of S456. The nipecotic acid derivatives in class C completely agree with the literature, thus further validating the binding hypothesis of selected hGAT1 antagonists.

On the basis of these findings, further analyses will focus on virtual screening followed by activity and ADME profiling of subsequent hits and optimization of respective chemical scaffolds structures to identify new arsenal of hGAT1 antagonists with improved efficacy and better ADME properties.

Conclusion

The current study probes the 3D structural features and binding hypothesis of hGAT1 inhibitors in the binding pocket of an in-house homology model of hGAT1 in the open-to-out conformation. Overall, our GRIND model illustrated the importance of two hydrogen bond acceptor groups at mutual distance of 8.00–8.40 Å, one hydrogen bond donor and one hydrophobic group at distance of 5.60–6.00 Å and 10.40–10.80 Å, respectively from one of the hydrogen bond acceptors for achieving high biological activity against hGAT1. However, the docking studies of nipecotic acid and N-diarylalkenyl piperidine analogs in the binding pocket of the hGAT1 model emphasized that the protonated nitrogen atom is oriented towards the extracellular side of the binding pocket due to the attachment of large hydrophobic moieties. In addition, Na1 and the residues G65, W68, Y140 and F294 in the binding pocket showed dominant interactions with the COOH group, aromatic moieties and the protonated nitrogen atom in the hGAT1 antagonists, respectively, that are important for achieving high activity against hGAT1. Both the GRIND model and the docking studies revealed that a meta-COOH group attached to the piperidine ring of hGAT1 antagonists is more favorable for interactions compared to an ortho or para substituted COOH group. Moreover, the hydrogen bonding and the specific shape/orientation of the antagonists were found to be significant for achieving highly potent hGAT1 antagonists. Overall, we anticipate that the current study may assist the development of more effective antagonists for the treatment of epilepsy and other associated neurological disorders.

Supplemental Information

Supplemental Information 1 General architecture of hGAT1 antagonists.

Click here for additional data file.

Supplemental Information 2 Schematic overview of 3D QSAR model training and statistical evaluation.

Click here for additional data file.

Supplemental Information 3 Brief overview of final cluster selection methodology for classes A, B and C.

Click here for additional data file.

Supplemental Information 4 Sequence alignment of hGAT1 (query is experimentally derived sequence taken from UniProt (ID: P30531) aligned with sequence of dopamine transporter (PDB: 4XP4) in Drosophila melanogaster. Identical/conserved residues (*), representing 66% similarity.

Click here for additional data file.

Supplemental Information 5 Quality assessment of hGAT1 model by ramachandran plot.

The most favored regions are marked as A, B and L whereas additionally allowed regions are indicated by a, b, l and p. Proline and non-glycine residues are marked as black squares. Filled tringles represent glycine residues.

Click here for additional data file.

Supplemental Information 6 Open-to-out conformation of hGAT1 model.

N-terminus and C-terminus residues are removed. Na+ ions are represented by blue spheres and Cl− ion by green sphere.

Click here for additional data file.

Supplemental Information 7 Correlation between biological activity (–logIC50) and GOLD score of selected poses from each classes A, B and C of the (A) top scored poses and (B) selected poses for analysis on the basis of ligand–protein interaction profiles.

Click here for additional data file.

Supplemental Information 8 The Ligplot analysis of hGAT1 interaction with compound 1 of class C.

Click here for additional data file.

Supplemental Information 9 Venn diagram represents overlapping and specific amino acids residues interacting with hGAT1 modulators in classes A, B and C, comparing with important amino acid residues known from the literature.

Click here for additional data file.

Supplemental Information 10 Database of hGAT1 antagonists along with biological activity values (IC50 µM).

Click here for additional data file.

Supplemental Information 11 Statistical parameters of PLS models developed by using multiple conformational sets at pH 7.4.

Click here for additional data file.

Supplemental Information 12 Test dataset, experimental vs. predicted −log IC50 and their respective residual values obtained via GRIND.

Click here for additional data file.

Supplemental Information 13 Correlation between the biological activities and top scored vs. selected poses on the basis of ligand–protein interaction profiles of compounds from classes A, B and C.

Click here for additional data file.

Additional Information and Declarations

Competing Interests

Author Contributions

Data Availability

The authors declare that the research was conducted in the absence of any commercial or financial relationships that could be construed as a potential conflict of interest.

Sadia Zafar performed the experiments, analyzed the data, contributed reagents/materials/analysis tools, prepared figures and/or tables, authored or reviewed drafts of the paper.

Ishrat Jabeen conceived and designed the experiments, analyzed the data, contributed reagents/materials/analysis tools, authored or reviewed drafts of the paper, approved the final draft.

The following information was supplied regarding data availability:

The data used in this work is available in the Supplemental Files.

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
