# Peer review of "GRID-independent molecular descriptor analysis and molecular docking studies to mimic the binding hypothesis of γ-aminobutyric acid transporter 1 (GAT1) inhibitors"

_PeerJ, doi:10.7717/peerj.6283_

## Round 0.1 · original submission · Major Revisions

Your manuscript has been revised by 3 reviewers and they have raised several points that you have to check or correct. I have also read the manuscript and I have the following questions/suggestions or corrections:

abstract

"The efficiency of this Na+ coupled GABA transport is regulated by an electrochemical gradient, which is directed inward under normal conditions. However, in certain pathophysiological situations, including strong depolarization or an imbalance in ion homeostasis, the GABA transfer may follow an increased rate of inward transport through GAT1." The sentences are affirming the same thing, thus, authors have to check the use "however" and have to clarify whether or not "under depolarizing conditions" the inward influx of GABA will be further increased in relation to the resting condition.

The statement: ". This mechanism may lead to the extra removal of extracellular GABA, which results in numerous neurological disorders including Alzheimer’s, schizophrenia, ..." in relation to AD and schizophrenia is too strong and is giving the impression that low extracellular GABA is causally linked with the diseases. I am not sure about that.

The same applies to the statements made in the introduction "...leads to several abnormal neurological disorders ...". Here the authors can change to "...which may be associated with abnormal neurological disorders..."

The sentence: "...Additionally, some reports also suggest that a photosensitizer precursor 5-aminolevulinic acid (5-Ala) used in photodynamic therapy may also inhibit the cellular uptake of GABA by GAT isoforms (Rud et al., 2000). ..." is a little misguiding. In fact, 5-ALA is an endogenous metabolite of Heme biosynthesis and to cite it as a precursor of a photosensitizer is less significant than cite its physiological role. Furthermore, old literature data have indicated that the toxicity of 5-ALA might be related to GABA imbalance. Anyway, the authors can simply delete the parts: "... that a photosensitizer precursor..." and "...used in photodynamic therapy...".

lines 83-94: It is not clear what was the template used for build the hGAT1 models. Was it the leucine transporter (LeuT) from Aquifex aeolicus (PDB ID: 3F3A) or the dopamine transporter from Drosophila melanogaster (dDAT, PDB ID: 4XP4)?

lines 103-104: Why antagonists with a molecular weight of less than 150 and IC50 >100 μM were removed? Please, change to "molecular mass" and check in Figure 1 (weight and change to "molecular mass" and check the signal >150)

line 194: Homology modeling. It is not clear what was the software used to make the protein models (Homology modeling).

lines 202-204: The values of sequence identity (46% and 25%), how it was obtained?

lines 115-116: Besides the Ramachandran plot, could be interesting the authors to use other software to validate the protein structure, such as Verify3D and ERRAT (http://servicesn.mbi.ucla.edu/).

lines 360-367: Was the ERRAT (http://servicesn.mbi.ucla.edu/ERRAT/) used in the protein validation? It is not clear. Please cite it in the lines 115-116.

line 356: Homology modeling of hGAT1. Please cite the percentage of residues in the most favored regions, additionally allowed regions, generously allowed regions and disallowed regions, obtained from the Ramachandran plot.

Figure 7 (pp. 37-38). Please add the respective interaction distances between the ligands and the hGAT1.

Table 3(pp. 47). Please verify in the Class B ligands if the -COOH group is drawn in the structure.

Please cite the program used in the protein sequence alignment (Figure S3).

The ligands present the -COOH group bound to chiral carbon, please discuss the stereochemistry (isomers R, S) of the molecules. How it affects in the binding pose? There is a preferred isomer?


Here in this part: "...Briefly, 300 2D as well as 3D descriptors available in MOE version 2013.0802 (MOE, 2013.08) were computed to obtain a distance calculation for each database entry. 20% of the data structures (19 compounds) with larger distance values from each other were selected as the test set and the remaining compounds (80%) were used to train the model (Minh et al., 2009). Additionally, a recently published data set of nipecotic acid derivatives were used as a validation set (Lutz et al., 2017)... ", the authors should add (as supplementary information) how the 80% remaining compounds were used to train the model.

Reviewer 1 ·

Basic reporting

The paper is well written, easy to follow and provides all raw data. Also, the statistics are shown and data (such as score values) are also presented as graphs to get a clearer overview on the data.Comprehensive and up to date literature is cited.

Authors address an important question from the viewpoint of developing GABA transport inhibitors. So far, only Tiagabine is the only FDA-approved drug against epilepsy, with a number of side effects, therefore the search for an effective inhibitor with no or minor side effects is eagerly awaited.

Experimental design

There have been many attempts to develop useful lead molecules or drug candidates as GABA inhibitors. GRID independent modeling is a relatively new approach in the field of SAR therefore, it was reasonable to apply the method for the study of GABA transport inhibitors.
The method was appropriate addressed, and it has been supplemented by molecular modeling and docking studies, as well.

Molecular modelling and docking was carried out appropriately. Modeling was based on the structure of dDAT, which was most probably the closest homologue of hGAT-1 at the time of the writing. (It is to note, that the structure of human DAT and SERT has been already determined, which will probably be even more similar templates than dDAT for further studies). Otherwise, altogether appropriate care was taken to the modeling process.

Concerning the GOLD docking procedure, a relatively large cavity was left for the antagonists to bind. This has the advantage of leaving enough space for the antagonists, however, it might have led to several binding modes in a number of cases.

The number of GA runs and other settings fit to the generally applied standards.
Methods were written carefully enough for reproduction.
.

Validity of the findings

The findings are nicely presented. Table 1 is especially carefully detailed, giving a nice example of how data is to be presented.

Overall, the findings are summarized in Fig. 9. Namely, two hydrogen bond acceptor contours (N1) at a distance of 8.00-8.40 Å from each other. and a hydrogen bond donor (O) at a distance of 5.60-6.00 Å, and finally a hydrophobic group at a distance of 10.40-10.80 Å form the hotspots for hGAT-1 inhibitors, therefore a COOH group at meta position of the piperidine ring may provide an important interaction point within the binding cavity of hGAT1.

This finding is well stated and linked to the original research question. It can be directly interpreted in terms of developing hGAT-1 - inhibitors in the future.

Additional comments

Overall, it is a well written paper, there are only a few questions remain to be answered.

In row 221 it is stated that the binding site was kept flexible.
However, in row 228 it is said that the side chains of hGAT-1 were kept rigid.
If you mean, they were kept rigid outside the binding site, please state it explicitly.

In supplementary Table 4 the top scoring hits get an unusually high score,(even around and over 70) while the score of the selected poses are much less and they cover a narrower range. Does it mean, that the top pose appeared only once, and there were several other poses? How many times did the "selected poses" appear in the run?

The quality of the figures 5, 6 and 7 should be improved, in terms of sizing. It may be due to the PDF processing, but the figures seem to be "pressed" from the sides. (It also appears in Table 2).

Reviewer 2 ·

Basic reporting

The paper is well written and very well organized.

About literature references I suggest the authors include some references of 2016, 2017 and 2018. There is only one paper of 2017 (Lutz et al. 2017) and all the other references are 2015 or older. For instance in Introduction, lines 73-88 that describes some related work should be updated as well as the description about new inhibitors for GAT1 (lines 67-72).

Some figures have parts distorted. Figure 6 B, the close view of class A, B and C are distorted and should be improved. Figure 7 should be in a better quality.

Experimental design

The research question of this paper is well defined and very relevant. The search for new inhibitors for GAT1 is an important topic of research, since according to the authors, there is only one second-generation FDA approved anticonvunsant agent for human sapies GAT1 (Tiagabine). Besides analogues of this drug Tiagabine that have been developed are often associated with off target toxicity and poor ADME-Tox properties (lead to side effects).

Some points of the methodology need to be more detailed to have sufficient information to be reproducible:
- the workflow of the hGAT1 antagonists data preprocessing: it is necessary to discuss about the chosen criterion (molecular weight and IC50) for filter the 375 non duplicated initial antagonists as well as the next steps “assay type” and “cell line” filters. On the text (lines 104-110) these steps are describe different from the figure ( in the text do not appear the number of 152 inhibitors, for instance).
- the validation set should be included on Figure 1.
- Improve the legends of the Figures. For instance the legend of Figure 2. How many molecules are on this figure?

The authors should discuss in Methodology the reasons they chose the applied tools (Pentacle v 1.06, AMANDA algorithm, CLACC algorithm. PSL analysis, GOLD, MOE, Modeller, Procheck and so on). For instance: for molecular docking, popular docking tools are Vina, GOLD, FleXX, but the authors opt to use GOLD for what reason?. The same for Modeller and so on.

Validity of the findings

The first point about the Results section is that part of its content should be moved to the Methods Section. I suggest the authors review the first paragraphs of results section because it is confuse. I believe that part of the lines 246-249, 251-255 should be on Section Methods.

Since part of the Results are related with the developed homology model I suggest the authors includes a figure of this model indicating the binding site through highlighting the main amino acids (for instance the protein in ribbons and these amino acids in lines).

In general the results are well described and discussed.

Additional comments

In this paper the authors present a study where various predictive GRIND models were developed to discover 3D features of hGATA1 inhibitors. Besides, the authors propose a binding hypothesis for the selected GAT1 inhibitors by molecular docking in the binding site of a human GAT1 homology model.
The paper is well written and very well organized however the Section Methods should be detailed to improve the general quality of the paper.

·

Basic reporting

The English language and all structural formats are acceptable.

Experimental design

The authors have done a good and credible study and they clearly explained the applied methods.

Validity of the findings

Data presentation and the results and discussion are concise and acceptable.

Additional comments

The manuscript organized and written very well. I recommend accepting in its current format.

---

## Round 0.2 · accepted · Accept

Thank you for revising your manuscript.

# Reviewer 1 ·

Basic reporting

No comment

Experimental design

No comment

Validity of the findings

No comment

Additional comments

Thank you for correcting the points I raised at the first review period. The paper is now ready for publication.

Reviewer 2 ·

Basic reporting

The literature references were improved on the reviewed version.
The authors also modified the Figures 6 and 7 as suggested.

Experimental design

The authors addressed some points indicated to improve on the Methods section.

Validity of the findings

The results are now well described.

Additional comments

The authors addresses almost all the points indicated on the first round of Review.